# T Cell-Induced Colitis Is Exacerbated by Prolonged Stress: A Comparison in Male and Female Mice

**DOI:** 10.3390/biomedicines12010214

**Published:** 2024-01-18

**Authors:** Ross M. Maltz, Pedro Marte-Ortiz, Madeline G. McClinchie, Miranda E. Hilt, Michael T. Bailey

**Affiliations:** 1Division of Pediatric Gastroenterology, Hepatology and Nutrition, Nationwide Children’s Hospital, Columbus, OH 43205, USA; 2Department of Pediatrics, The Ohio State Wexner Medical Center, Columbus, OH 43210, USA; 3The Center for Microbial Pathogenesis, The Research Institute, Nationwide Children’s Hospital, Columbus, OH 43205, USA; 4Oral and Gastrointestinal Microbiology Research Affinity Group, Abigail Wexner Research Institute, Nationwide Children’s Hospital, Columbus, OH 43205, USA; 5Biomedical Sciences Graduate Program, Wexner Medical Center, The Ohio State University, Columbus, OH 43210, USA

**Keywords:** chronic T cell-mediated colitis 1, inflammatory bowel disease 2, sex 3, chronic stress 4

## Abstract

Psychological stress exposure is well recognized to exacerbate inflammatory bowel disease (IBD) but the mechanisms involved remain poorly understood. In this study, chronic T cell-mediated colitis was induced by adoptively transferring CD4^+^CD45RB^high^ splenic T cells from C57BL/6 WT donor mice into *Rag1^tm1Mom^* mice. Two weeks after T cell transfer, mice were exposed to a prolonged restraint stressor (RST) for 8 h per day for 6 consecutive days. The colitis phenotype was assessed via histopathology and semi-quantitative rt-PCR at humane endpoints or 10 weeks post-T-cell transfer. Mice that received the T cell transplant developed chronic colitis marked by increases in colonic histopathology and inflammatory cytokines. Colonic histopathology was greater in males than females regardless of RST exposure but RST exposure increased histopathology scores in females such that they reached scores observed in the males. This pattern was consistent with cytokine gene expression and protein levels in the colon (especially for IFN-γ, IL-17A, and TNF-α). Serum cytokine levels were not strongly affected by exposure to the stressor. Using a murine model of chronic T cell-mediated colitis, this study demonstrates that biological sex strongly influences colonic inflammation and exposure to chronic stress has a more pronounced effect in females than in males.

## 1. Introduction

Inflammatory bowel disease (IBD), which refers to both ulcerative colitis (UC) and Crohn’s disease (CD), is a chronic intestinal inflammatory disorder that impacts more than three million individuals in the United States [1]. It is thought that IBD develops in patients with a genetic predisposition when an aberrant immune response to the commensal gut microbiome and environmental triggers such as stressor exposure activates the expansion of T cells, leading to the production of pro-inflammatory cytokines and the infiltration of leukocytes such as monocytes and neutrophils that result in chronic inflammation, goblet cell depletion, and epithelial cell injury [2,3,4,5,6].

Inflammatory bowel disease affects men and women at similar rates but there are sex-specific trends in disease progression that are confounded by age and geographic area [7]. Whether males and females with IBD are equally affected by stress is not well understood. However, sex hormones (estrogens, progesterone, and androgen) can be affected by stress [8], impact the regulation of T cell responses [9,10], and are known to play an influential role in the physiology and pathophysiology of the intestine [9,10,11]. Prior studies have suggested that progesterone and estrogen can reduce inflammatory cytokines and can improve epithelial barrier function and tight junctions [12]. Other studies have implicated the estrogen receptor beta (ER-β) in estrogen-mediated changes to gastrointestinal physiology [13]. In humans, decreased ER-β expression and IL-6 dysregulation were observed in the colonic mucosa in patients with active CD and UC [10]. Interestingly, ER-β deficiency in mice enhances dextran sodium salt (DSS)-induced colitis through effects on the microbiome [14]. This is consistent with studies suggesting that sex-based differences in the microbiome may contribute to sex-based differences in inflammatory diseases [11,15,16]. The mechanisms through which this occurs are not well understood and sex-based differences in the microbiome have not been completely delineated. Some studies have found that men have lower microbial diversity than women, while others have found no difference in alpha diversity, but a significant difference in beta diversity between men and women [12,17,18]. Interestingly, taxa that have been suggested to influence colonic inflammation (such as *Akkermansia muciniphila* and *Bacteroides* spp.) have been shown to be differentially abundant in males and females [17,19] but whether this is related to differences in intestinal inflammation is not yet known.

Individuals with IBD experience a variety of psychological stressors and stress has been associated with the pathogenesis of IBD [20,21]. Additionally, those diagnosed with IBD have higher rates of depression and anxiety [22]. However, our understanding of the relationship between stress and IBD remains unclear and whether stress has different effects in males and females is not yet known. Animal models of stress have provided more conclusive evidence of the relationship between stress and gastrointestinal inflammation [23,24]. Specifically, exposure to a prolonged restraint stressor (RST) has been shown to alter the composition of the gut microbiome (including reductions in alpha diversity and *Lactobacillus* spp. relative abundances [25,26]), affect short-chain fatty acids, increase corticosterone, and lead to increased colonic inflammation, including inflammatory cytokines and chemokines, such as IL-1β, TNF-α, IL-17A, IL-6, and CCL2, after pathogen exposure [25,26,27]. Thus, this study used prolonged RST to test whether stressor exposure would exacerbate colonic inflammation in male and female mice.

T-cell transfer is a particularly advantageous model of colitis because it allows for the investigation of the immunological events that initiate and perpetuate disease [28]. The T cell transfer model involves the adoptive transfer of CD4^+^CD45RB^high^ T cells from wild-type (WT) mice into a syngeneic immune-deficient mice [3,6]. Within 6 to 8 weeks, the T cell transfer will induce moderate to severe chronic colitis affecting both the small and large intestines. Disease manifestations include hunching, progressive weight loss, and diarrhea [3,4,5,6]. Histopathology shows a marked decrease in goblet cells, myeloid cells (neutrophils and monocytes), and lymphocyte infiltration and, in severe cases, crypt abscessation [3,4,5,6]. Additionally, the recipient mice show substantial increases in tumor necrosis factor (TNF)-α, IFN-γ, IL-1β, and other inflammatory cytokines associated with a Th1/Th17 immune response [3,4,5]. We used the T cell transfer model of colitis in male and female mice that were subsequently exposed to the RST stressor to test whether stress has a similar effect on colonic inflammation in male and female mice.

## 2. Materials and Methods

### 2.1. Animals

Male and female Rag1^tm1Mom^ mice on a C57BL/6J background were obtained from Jackson Laboratories (Bar Harbor, ME, USA) at 6–8 weeks of age and bred at The Research Institute at Nationwide Children’s Hospital. Pups were weaned at 21 days and housed 3 per cage by sex. Male and female C57BL/6 wild-type (WT) mice that were 8 weeks old were purchased from Charles River Laboratories (Wilmington, MA, USA). All mice were maintained at the Animal Resource Core facility in a controlled room (temperature 20–25 °C, humidity 30–70%, and 12-h light-dark cycle) with ad libitum access to standard chow and water. All experiments were approved by the Institutional Animal Care and Use Committee at The Research Institute at Nationwide Children’s Hospital (Columbus, OH, USA) protocol number AR16-00091.

### 2.2. Induction of Chronic Colitis

Using a well-established protocol [3], splenic CD4^+^CD45RB^high^ T cells from C57BL/6 WT mice were adoptively transferred into same-sex 8–12-week-old Rag1^tm1Mom^ mice. First, the spleens from C57BL/6 WT mice were removed and placed in FACS buffer (PBS, fetal bovine serum, and sodium azide). Then, the spleens were homogenized by rubbing between glass slides until a single-cell suspension was created. The single-cell suspension was passed through a 70µm cell strainer. Cells were enriched by negative isolation using Dynabeads Untouched Mouse CD4 Cell Kit (Life Technologies AS, Carksbad, CA, USA) according to the manufacturer’s instructions. CD4 cells were fluorescently labeled with fluorescein isothiocyanate (FITC) rat anti-mouse CD45RB (Thermo Fischer Scientific, Waltham, MA, USA) and phycoerythrin (PE) rat anti-mouse CD4 (BD Biosciences, Franklin Lakes, NJ, USA). CD4+ T cells were sorted using a Becton-Dickenson Influx cell sorter by double gating on CD4^+^ singlet cells and the brightest 40% CD45RB^high^ cells. Cells were sorted into tubes containing post-sorting buffer (PBS, EDTA, FBS) to stabilize the post-sorted T cells. In total, 5 × 10^5^ CD4^+^CD45RB^high^ T cells were then injected intraperitoneally into recipient Rag1^tm1Mom^ mice of the same sex as the T cell donors.

### 2.3. Experimental Design

The objective of the experiment was to evaluate whether exposure to stress after T cell transfer would enhance T cell-induced colitis in male and female mice. A total of 90 mice were divided into 12 different experimental rounds (6 male groups and 6 female groups). Half of the Rag1^tm1Mom^ mice were recipients of CD4^+^CD45RB^high^ T cells (9 mice per group) and the other half were mice that did not receive T cells (6 mice per group). One-third of the mice from each group were exposed to a prolonged restraint (RST) stressor for 6 consecutive days (Figure 1A). The RST stressor was conducted by placing the mice in well-ventilated 50 mL conical tubes for 8 h from 1000 to 1800. Mice exposed to the RST stressor will not eat or drink while in the tubes, even if food and water are provided. Thus, to account for the food and water deprivation in the RST group, a control group was added that was deprived of food and water (FWD group) for 8 h a day on 6 consecutive days but was not exposed to RST. A second control group (designated home cage control (HCC)) had food and water available ad libitum and was not exposed to RST.

The RST stressor and FWD were started 14 days after T cell transfer (Figure 1B). All mice were weighed and monitored for changes in appearance, stool consistency, and blood in the stool. Mice were euthanized if they lost more than 15% of their body weight or were lethargic and failed to move when gently prodded. Otherwise, all mice were euthanized 10 weeks after receiving T cells.

### 2.4. Histopathology

At each endpoint, the colons were harvested and the distal colons were fixed in 10% formalin-buffered phosphate, embedded in paraffin, and stained with hematoxylin and eosin (H&E) for blinded histopathology scoring using a validated scoring system [29] under 400X magnification (Zeiss ZX10 microscope, Oberkochen, Germany). A score of 0 represented no inflammation; 1 indicated mild inflammation characterized by a few multifocal mononuclear cell infiltrates in the lamina propria with minimal epithelial hyperplasia and no loss of mucus from goblet cells; 2 marked moderate inflammation represented with several multifocal cell infiltrates in the lamina propria, mucin depletion, small epithelial erosions, and mild epithelial hyperplasia; 3 indicated moderate/severe inflammation represented with submucosa inflammation, crypt abscesses, mucin depletion, and moderate epithelial hyperplasia; and a score of 4 represented severe inflammation with lamina propria infiltration, architectural distortion, crypt abscesses, and ulcers [29].

### 2.5. Semiquantitative Real-Time PCR

Mid colons were snap frozen in liquid nitrogen and stored at −80 °C until isolating total RNA using Tri-zol reagent (Invitrogen, Carlsbad, CA, USA) following manufacturer protocols. Complimentary DNA was synthesized with the Avian Myeloblastosis Virus (AMV) Reverse Transcriptase kit (Promega Corporation, Madison, WI, USA). All mouse mRNA primers were purchased from Integrated DNA Technologies (Redwood City, CA, USA). Real-time PCR was conducted using the QuantStudio 3 system (Applied Biosystems, Bedford, MA, USA). The housekeeping gene Eef2 was used and the relative amount of transcript was determined using the comparative cycle threshold (C_t_) method as previously described [26,30]. Primer sequences are provided in Appendix A. Data were expressed as a fold change from the female HCC group that received T cells because on average, this group had the lowest amount of histopathology.

### 2.6. Serum and Colonic Measurement of Cytokines

Blood was collected via cardiac puncture at the time of tissue collection and serum was isolated via centrifugation. Protein was isolated from frozen colonic tissue by first homogenizing tissue with a mortar and pestle in a dry ice bath for 20 min. After homogenization in the mortar and pestle, frozen tissue was placed into a 2 mL screw cap tube containing a 2.8 mm ceramic bead and 500 µL of lysis buffer (RIPA lysis buffer and Extraction BufferHalt Phosphatase Inhibitor Cocktail (Thermo Scientific, Wlatham, MA, USA)). The tubes were placed on ice for 10 min and then transferred to an Omni Bead Ruptor 4 (Omni Internation, Castle Rock, CO, USA) at full speed for 1 min. Homogenized samples were centrifuged at 14,000 RPM for 15 min at 4 °C. Supernatants were used in downstream assays. Cytokine expression in serum and proteins isolated from distal colonic tissue was measured using a U-plex T cell combo kit. This kit simultaneously measures 14 cytokines/chemokines (i.e., GM-CSF, IFN-γ, IL-2, IL-4, IL-9, IL-10, IL-13, IL-17A, IL-17E/IL-25, IL-17F, IL-21, IL-22, MIP-3α (a.k.a. CCL20), and TNF-α; Meso Scale Diagnostics (Rockville, MD, USA)). The plates were analyzed on the MESO QuickPlex SQ 120 imager (Meso Scale Discovery, Rockville, MD, USA). Cytokine concentrations were calculated by a 4-parameter logistic non-linear regression analysis of the standard curve. Cytokine levels were expressed as pg/mL for serum and were adjusted to pg/mg of colon tissue weight.

### 2.7. Statistical Analysis

A two-way ANOVA was used to analyze histopathology scores, colonic cytokine gene expression, colonic cytokine levels, and serum cytokine levels with stressor exposure (HCC, FWD, and RST) and sex (male vs. female) as the between subject’s variables. Statistical outliers were removed from the data set based on the criteria of greater than two standard deviations from the mean. An alpha level of *p* < 0.05 was set as the rejection criteria for the null hypothesis. All data were analyzed using SPSS statistical software version 26 (IBM Corp, Armonk, NY, USA).

## 3. Results

### 3.1. RST Does Not Lead to Colitis in the Absence of T Cell Transfer in Male or Female Rag1^tm1Mom^ Mice

None of the male or female *Rag1^tm1Mom^* mice that did not receive an intraperitoneal injection of splenic CD4^+^CD45RB^high^ T cells from C57BL/6 WT donor mice but which were exposed to RST (as well as the FWD or HCC control conditions) showed evidence of sickness or colitis. These mice continued to gain weight through the experiment (Figure 2A) and at the end of the 10-week experiment, these mice did not have any differences in colonic histopathology (Figure 2B). Since we were interested in the effects of restraint stress on T cell transfer colitis, data from these sex-matched vehicle controls were not included here. In contrast, male and female Rag1^tm1Mom^ mice that received an intraperitoneal injection of splenic CD4^+^CD45RB^high^ T cells from C57BL/6 WT donor mice developed a colitis phenotype, characterized by colonic histopathology and increased expression of inflammatory cytokines in the colon and serum.

### 3.2. Male Mice Lose Significantly More Weight Than Female Mice Regardless of RST Exposure

Mice were weighed regularly throughout the experiment and at the time of tissue collection. Male mice lost weight, whereas female mice gained weight at the time of tissue collection independently of RST exposure, resulting in sex being a significant main effect (*p* < 0.05; Figure 2C). Interestingly, the weight change was less pronounced in the RST groups compared to the FWD or HCC controls in both males and females. Female mice exposed to RST gained the least amount of weight (Figure 2A,C).

### 3.3. Sex and RST Exposure Lead to Significantly Altered Colonic Cytokine Expression and Histopathology

We tested whether cytokine expression in the middle colon was affected by exposure to the RST stressor in male and female mice (Figure 3). The expression of genes related to Th1 and Th17/Th22 pathways was significantly different in mice exposed to stress. Specifically, there were statistically significant main effects of stress for Tnfa, Ifng, Il1b, Il2, and Il17a (*p* < 0.05; Figure 3A–E). Although these stress effects were most evident in the female mice, there were no statistically significant sex x stress interaction effects in the ANOVAs. However, there were statistically significant sex main effects. Overall, male mice had significantly higher expression of Ifng, Il1b, Il2, Il17a, Il22, Il7, Il6, and Nos2 (*p* < 0.05; Figure 3B–I). Consistent with the differences in cytokine gene expression, colonic histopathology scores were significantly higher in stressor-exposed mice compared to non-stress control mice (main effect of stress, *p* < 0.05) and in male mice compared to female mice (main effect of sex, *p* < 0.05; Figure 3K–M).

### 3.4. RST Exposure and Sex Significantly Alter Cytokine Protein Levels in the Colon

The differences in colonic gene expression led to assessment of cytokine protein levels in the mid colon. Exposure to stress led to significantly higher levels of the cytokines IL-2, IL-17A, and IL-22 (main effect of group, *p* < 0.05; Figure 4C–E). Post-hoc tests indicated that IL-2 was higher in RST compared to FWD but not HCC mice (*p* < 0.05; Figure 4C), whereas IL-17A was higher in RST compared to both HCC and FWD groups (*p* < 0.05; Figure 4D,E). In many cases, the effects of stress seemed to be strongest in female mice; however, there was only one sex x stress interaction in the ANOVA’s, showing that stress increased TNF-α in female mice (RST vs. HCC in females, *p* < 0.05; RST vs. HCC in males, *p* > 0.05). In addition, TNF-α was higher in male HCC vs. female HCC (*p* < 0.05; Figure 4A). Despite the lack of significant sex x stress interactions for the other cytokines, there were multiple main effects of sex, indicating that male mice had significantly higher levels of IFN-γ, IL-13, and GM-CSF (main effects of sex, *p* < 0.05; Figure 4B,H,M). The cytokines IL-17E/IL-25, IL-13, IL-4, IL-10, IL-9, and CCL20 were not different in stressor exposed mice (compared to non-stress control mice) and did not differ in males vs. females (Figure 4G–L).

### 3.5. RST and Sex Significantly Alter Cytokine Levels in Serum

Cytokine levels tended to be lower in the serum compared to the colon and IL-21 and IL-9 levels were only detectable in a portion of the mice (16/42 and 15/42). Nonetheless, serum cytokine levels were assessed. Exposure to RST led to a significant increase in IL-17E/IL-25 in both males and females (main effect of stress, *p* < 0.05; Table 1). Additionally, male mice had higher levels of CCL20 compared to females (main effect of sex, *p* < 0.05; Table 1). No other statistically significant difference in cytokine or chemokine expression was observed between the RST-exposed mice and the non-stress control mice or between males and females (Table 1).

## 4. Discussion

Our study aimed to investigate whether exposure to a chronic RST impacts intestinal inflammation in the T cell transfer colitis model and found that it significantly increases cytokines in the colon of both male and female mice with T cell-induced colitis but that the observed increases were more substantial and widespread in females compared to males. Additionally, we found that males with T cell-induced colitis have higher baseline levels of colonic inflammation than females with T cell-induced colitis, as shown by their colonic histopathology scores and cytokine expression of controls. Moreover, RST exposure significantly increases colonic histopathology scores in females but this is less evident in males with T cell-induced colitis. This is likely due to the finding that males had more colonic histopathology than females regardless of whether they were exposed to stress, thus making stress-induced increases less evident in the males. Overall, our results suggest that RST exposure exacerbates T cell-induced colitis in females but, in general, males have more colitis than females.

Reduced body weight is a common consequence of cytokine-driven sickness in mice [31]. The expansion of cytokines during sickness causes disruptions to homeostatic conditions, resulting in increased body temperature, increased sleep, reduced appetite and accompanying weight loss, and changes in metabolism [32]. Weight loss is a common symptom of the inflammatory response and has been used as an indicator of cytokine-driven sickness in mice [32,33]. The higher inflammation in male mice compared to female mice was consistent with our finding that male mice lost weight, whereas female mice gained weight at the time of tissue collection independent of RST exposure. The RST female mice gained the least amount of weight, which is consistent with the increased colitis in the female mice exposed to RST.

Sex differences have previously been identified in experimentally induced murine colitis; however, there are conflicting results and conclusions on the influence of sex on disease severity [25,34]. In one study using DSS-induced colitis, researchers found that female mice had less colonic inflammation, indicated by longer colon length, less inflammatory infiltrates, and less crypt damage [35]. In a different study using IL-10^−/−^ mice, which develop spontaneous enterocolitis, researchers found that female mice had a shorter colon length and greater levels of IL6 and TGFβ in the colon compared to males [36]. Additionally, Goodman et al. (2014) showed female mice displayed onset of ileitis earlier with a more severe phenotype than male mice in the chronic SAMP1/YitFC model of Crohn’s disease [37]. Our results show that female mice with T cell-induced colitis alone have significantly less colonic tissue damage and lower inflammatory cytokine expression than males with T cell-induced colitis alone. However, in our study, when female mice were exposed to RST, the T cell-induced colitis was significantly increased, as shown by the significantly increased colonic histopathology scores, cytokine gene expression, and cytokine protein levels. It must be noted that all of these studies utilized different models of IBD. Thus, additional studies are needed to identify which components of the models differ in males and females and how this may be affected by stress.

Prior research has also highlighted sex-based differences in IBD pathogenesis [3,26,29], disease type [3,29], and treatment [3,26,29,30,38] in humans but with similarly conflicting results. UC and CD affect men and women at similar rates and sex-specific trends in relative risk are further confounded by age and geographic area. Females with UC and CD have more frequent skin, joint, and eye extraintestinal manifestations (EIMs) whereas males more frequently develop osteopenia/osteoporosis, sclerosing cholangitis, and ankylosing spondylitis [7,39]. One study found that more male CD patients underwent small bowel and ileocecal resection than female patients [39]. Biological sex also influences therapeutic outcomes [7,40]. One study found that more male CD patients were treated with prednisone than female CD patients [39]. Females more frequently experience side effects to anti-TNF therapies, while males are more likely to experience a loss of response to anti-TNF therapies and require increased dosage [7].

Stress is well recognized to affect IBD severity and symptoms but whether this occurs equally in males and females is not well studied. Our study in mice suggests stress affects colitis more strongly in females compared to males but similar sex comparisons need to be made in human studies. However, sex differences in stress related conditions, such as anxiety, have been studied in people with IBD. Individuals with IBD report higher levels of anxiety and depression compared to the general population [13,30,41,42] and patients with active IBD experience higher rates of anxiety and depression [41,43]. Females with IBD have higher rates of anxiety [13,44] compared to males with IBD.

## 5. Conclusions

This study used the T cell transfer murine model of IBD to investigate the effects of psychological stress on both colonic and systemic inflammation in male and female mice. We found that T cell-induced colitis was greater in males compared to females in terms of higher inflammatory cytokine gene expression and protein levels, as well as higher colonic histopathology scores. Exposure to the RST stressor caused a significant increase in colonic inflammation in both male and female mice with T cell colitis but this was most evident in females due to the low levels of colitis in the female HCC controls. Our results suggest that T cell transfer may induce a milder colitis in female mice but they are more susceptible to stress-induced exacerbation of colonic inflammation.

## Figures and Tables

**Figure 1 biomedicines-12-00214-f001:**
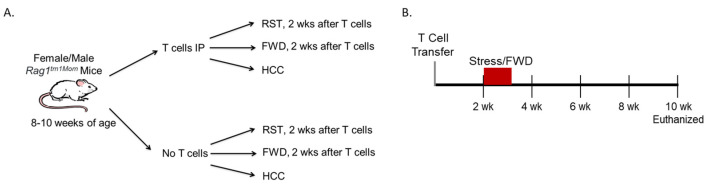
Experimental design. (**A**) Outline showing the assignment of mice used in this study. (**B**) Schematic showing the timing of T cell transfer, stressor exposure, and tissue collection.

**Figure 2 biomedicines-12-00214-f002:**
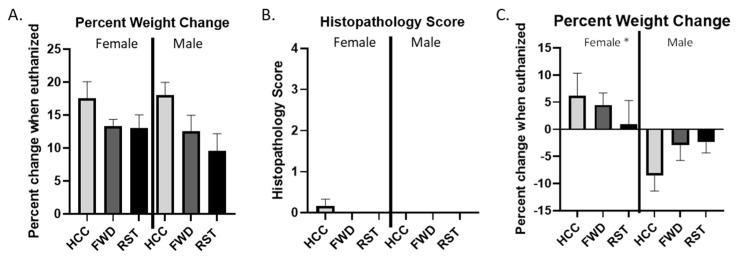
T-cell transfer is necessary for sickness-related weight changes. (**A**) Female and male Rag1^tm1Mom^ that did not receive T cells from C57BL/6 donor mice gained weight over the course of the experiment and (**B**) showed no evidence of colonic histopathology. (**C**) Male Rag1^tm1Mom^ mice that received T cells from C57BL/6 donor mice lost weight over the course of the experiment whereas female Rag1^tm1Mom^ that received T cells from C57BL/6 donor mice gained weight over the course of the experiment. n = 9 mice per sex, per group. * main effect of sex, *p* < 0.05.

**Figure 3 biomedicines-12-00214-f003:**
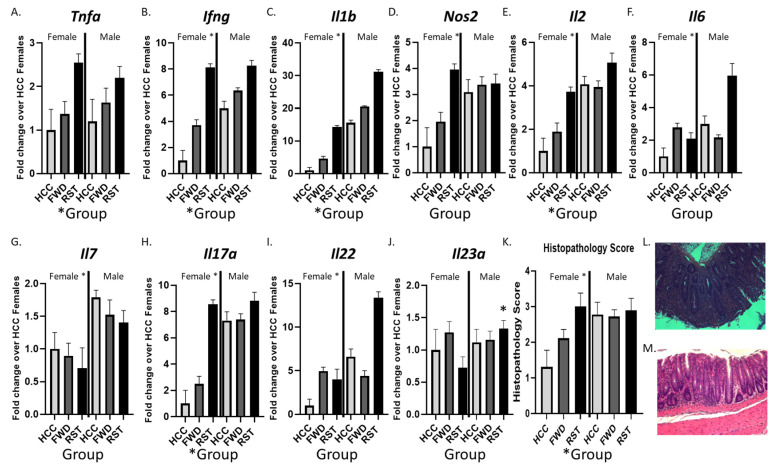
Colonic cytokine gene expression and histopathology scores differ in males and females exposed to RST or HCC conditions. (**A**–**J**) The mid colon was collected and gene expression in female and male Rag1^tm1Mom^ that received T cells from C57BL/6 donor mice prior to exposure to RST, FWD, or HCC control conditions was assessed using real-time PCR. (**K**) The distal colon was collected and fixed in 10% formalin-buffered phosphate for H&E staining prior to histopathology scoring. (**L**) Representative image of H&E stained section of the distal colon from a female Rag1^tm1Mom^ that received T cells from C57BL/6 donor mice prior to exposure to RST. (**M**) Representative image of an H&E stained section of the distal colon from a female Rag1^tm1Mom^ that received T cells from C57BL/6 donor prior to the FWD control condition. Female * indicates sex having a significant main effect (*p* < 0.05). * Group indicates significant main effect of group (*p* < 0.05). * Over a bar indicates significant sex x group interaction followed by protected t-test as a post-hoc test (*p* < 0.05). n = 9 per group, per sex.

**Figure 4 biomedicines-12-00214-f004:**
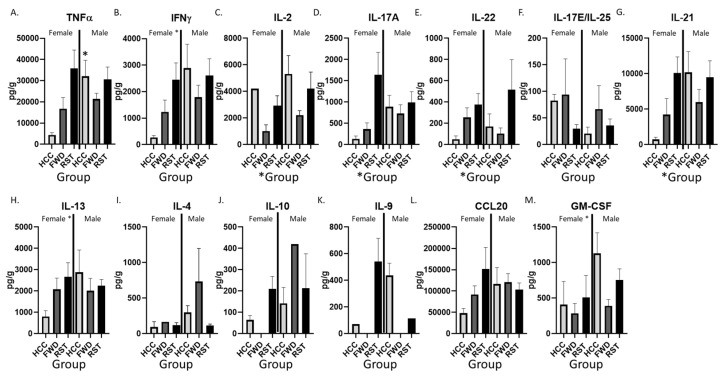
Colonic cytokine levels differ in males and females exposed to RST or HCC conditions. (**A**–**M**) The distal colon was collected and protein extracted to measure cytokine levels using the U-Plex T cell combo kit (Meso Scale Diagnostics, Rockvillem MD, USA). Female * indicates significant main effect of sex (*p* < 0.05). * Group indicates significant main effect of group (*p* < 0.05). * Over a bar indicates significant sex x group interaction followed by protected *t*-test as a post-hoc test (*p* < 0.05). n = 9 per group, per sex.

**Table 1 biomedicines-12-00214-t001:** Serum cytokine levels in male and female mice with T cell-mediated chronic colitis.

Cytokines	HCC Female	FWD Female	RST Female	HCC Male	FWD Male	RST Male
TNFα	416 ± 90.6 (6)	690.8 ± 189.4 (7)	1122.2 ± 254.1 (7)	758.3 ± 195.3 (7)	1087.8 ± 138.6 (7)	878.9 ± 100.2 (7)
IFNγ	134.1 ± 54.3 (6)	290.1 ±105.0 (7)	237.9 ± 52.9 (7)	257.3 ± 82.5 (7)	316.5 ± 37.9 (7)	305.5 ± 45.0 (7)
IL-2	126.9 ± 39.1 (7)	97.9 ± 14.7 (7)	112.2 ± 22.5 (7)	112.1 ± 41.9 (7)	136.9 ± 13.3 (7)	157.5 ± 10.6 (7)
IL-17A	56.0 ± 16.9 (7)	66.9 ± 14.0 (7)	97.9 ± 28.1 (7)	106.1 ± 30.4 (7)	106.2 ± 20.4 (7)	103.1 ± 19.0 (7)
IL-22	258.8 ± 57.8 (7)	283.1 ± 109.2 (7)	698.0 ± 263.4 (7)	308.4 ± 136.2 (7)	394.6 ± 109.1 (7)	368.1 ± 76.6 (7)
IL-21	13.1 (1)	33.9 (1)	22.4 ± 5.3 (5)	34.0 ± 19.1 (3)	8.8 ± 0.9 (2)	14.8 ± 5.0 (3)
IL-17E/IL-25	2.4 ± 1.1 (7)	4.9 ± 2.4 (7)	5.9 ± 1.8 (7)	3.4 ± 0.7 (7)	2.8 ± 0.9 (7)	5.2 ± 0.7 (7)
IL-17F	200.9 ± 73.7 (4)	59.8 ± 46.8 (2)	188.1 ± 49.3 (5)	106.5 ± 34.8 (2)	92.2 ± 42.8 (2)	120.8 ± 103.9 (4)
IL-13	68.8 ± 27.2 (4)	48.4 ± 33.5 (2)	110.1 ± 14.2 (6)	78.8 ± 23.3 (3)	67.7 ± 12.0 (6)	72.4 ± 12.9 (5)
IL-4	13.5 ± 3.5 (7)	9.9 ± 4.4 (7)	10.3 ± 3.8 (7)	6.9 ± 3.5 (7)	11.0 ± 1.9 (7)	11.0 ± 4.5 (6)
IL-10	140.6 ± 20.4 (7)	150.7 ± 29.2 (7)	165.6 ± 17.3 (7)	129.2 ±30.9 (7)	139.7 ± 11.9 (7)	120.7 ± 8.3 (7)
IL-9	NDL (0)	19.7 ± 16.1 (2)	31.0 ± 8.6 (4)	39.3 ± 0.6 (2)	21.2 ± 11.5 (4)	14.5 ± 4.2 (4)
CCL20	2178.6 ± 493.3 (7)	2716.6 ± 605.8 (7)	4473.8 ± 1212.2 (6)	2946 ± 802.5 (6)	9600 ± 3183.6 (7)	11,045 ± 3883.5 (6)
GM-CSF	3.8 ± 1.2 (7)	3.5 ± 0.8 (7)	3.7 ± 0.8 (7)	3.6 ± 1.0 (7)	4.3 ±0.4 (7)	4.9 ± 1.3 (7)

Note: Data are the mean ± SEM of the pg/mL for each cytokine. The number in () indicates the number of samples that had detectable levels of each cytokine. Cytokines were measured in n = 7 samples per group, per sex.

## Data Availability

The data presented in this study are available on request from the corresponding author. The data are not publicly available due to data sharing limitations.

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
