# Peer review of "T Cell-Induced Colitis Is Exacerbated by Prolonged Stress: A Comparison in Male and Female Mice"

_biomedicines, 2024, doi:10.3390/biomedicines12010214_

Round 1

Reviewer 1 Report

Comments and Suggestions for Authors

1. Does the gender difference in donor mice lead to graft rejection in an IBD model induced by CD4+CD45RB+T cell adoption affect the degree of inflammation in the IBD model? 

2. For female mice, the inflammatory factors IL17a and IFN gamma (fig3) that have been well studied in the IBD model show significant changes, but there is no significant change in body weight (fig1), and the histological scores are not consistent (fig2k); While detecting the expression of inflammatory factors in tissues and serum, it is recommended to measure the effect of RST on the expression of sex hormones.

Author Response

Reviewer 1 Comments:

  1. Does the gender difference in donor mice lead to graft rejection in an IBD model induced by CD4+CD45RB+T cell adoption affect the degree of inflammation in the IBD model?

Response: In our experimental model, the donor C57BL/6 mice were sex-matched to the recipient Rag1tm1mom mice. We did not examine whether gender differences in the donor mice can lead to enhanced graft rejection/colitis, but this is indeed an interesting question for the future (i.e., do sex differences in the T cells contribute to the enhanced colitis?).  An additional interesting question is whether stress effects on the T cells contribute to the increased colitis. In this regard, we have studies planned to adoptively transfer T cells from stressor-exposed mice into non-stress Rag1tm1mom mice.  Although these are interesting questions, these studies were beyond the scope of the current manuscript.

  1. For female mice, the inflammatory factors IL17a and IFN gamma (fig3) that have been well studied in the IBD model show significant changes, but there is no significant change in body weight (fig1), and the histological scores are not consistent (fig2k); While detecting the expression of inflammatory factors in tissues and serum, it is recommended to measure the effect of RST on the expression of sex hormones.

Response: We agree with the reviewer that IL17a and IFN-γ have been well studied in IBD models, including this T cell transfer model. While Il17a gene expression (Fig. 3H) closely tracks with histopathology scores (Fig. 3K), Ifng gene expression (Fig. 3B) does not track as well with histopathology scores. However, when protein levels, instead of gene expression, are assessed, the patterns of differences in both IL-17a (Fig. 4D) and IFN-γ (Fig. 4B) protein match the histopathology score patterns. In particular, IFN-γ protein is high in females exposed to RST and males exposed to HCC, FWD, or RST conditions; IL-17a is highest in females exposed to RST (albeit with high variability), and IL-17a is high in males exposed to HCC, FWD, or RST.  This pattern is similar to the pattern observed for histopathology scores.

The reviewer is correct, that the body weight differences do not track well with colitis scores (or colonic cytokines or chemokines).  We initially thought that differences in body weight would correlate with differences in serum cytokines. However, there were few differences in serum cytokines and serum cytokine levels did not correlate with body weight.  Unfortunately, we did not measure the effects of RST on the expression of sex hormones. Previous studies have demonstrated that progesterone increases with restraint stress equally in male and female rats (e.g., Kalil et al., 2013), but this study did not assess mice, nor did it assess longer/repeated bouts of restraint. Stress is well known to have broad, inhibitory effects on the hypothalamic-pituitary-gonadal axis, with a down regulation of hypothalamic gonadotropin-releasing hormone, which is responsible for the stimulation of follicle stimulating hormone and lutenizing hormone. These hormones, in turn, regulate steroid hormone production, thus leading to reductions in estrogens (Joseph et al., 2017). Interestingly estrogen has been suggested to have a protective role in a colitis murine model through activation of Gpr30 (Fidya et al., 2023). In addition, activation of estrogen receptor β has been shown to alleviate inflammatory lesions in a rat model of IBD (Jiang et al., 2021) further supporting that estrogens may be protective against colonic inflammation.  Thus, RST-induced reductions in estrogen could have accounted for the increased colitis severity in RST-exposed females. 

Kalil, B., et al. (2013). "Role of sex steroids in progesterone and corticosterone response to acute restraint stress in rats: sex differences." Stress 16(4): 452-460).

Joseph D.N. and Whirledge S. (2017). “Stress and the HPA Axis: Balancing homeostasis and fertility.” International Journal of Molecular Sciences, 18: 2224.

Fidya, C. N., et al. (2023). “Protective role of estrogen through G-protein coupled receptor 30 in a colitis mouse model.” Histochm Cell Biol. Doi: 10.1007/s00418-023-02235-z (epub ahead of print).

Jiang Q., et al. (2021). “Estrogen receptor β alleviates inflammatory lesions in a rat model of inflammatory bowel disease via down-regulating P2X7R expression in macrophages.” Int J Biochem Cell Biol. 139:10608

Reviewer 2 Report

Comments and Suggestions for Authors

Dear author, the submitted manuscript contains very interesting information on the subject, however, it is suggested that the following aspects be considered

Introduction section

It is suggested to integrate information on the diversity of the microbiota described in lines 55-56.

Could you indicate what you mean by evidence of the relationship between stress and intestinal inflammation (line 64-64), do you refer to some type of biomarkers? If so, could you integrate the information?

According to this information indicated in line 66, which phyla or genera of the intestinal microbiota were modified? 

Methodological section

In the histology section, it is necessary to include information on the microscope with which the samples were observed, as well as to indicate on the basis of which author the classification of the damage was established.

It is necessary to eliminate the information of line 134 because it should be a figure caption and this information is repeated 

Figure 1: Experimental design showing that T cell transfer is necessary for sickness-related weight changes". 

At the quantification of cytokines it is necessary to include the concentration of protein that was used to determine the different cytokines. 

Results Section

It is suggested to modify figure 1, because in this one the information of the biological model and results are being mixed, in this sense they could be separated, one figure could be only the one of the model, that is to say clause a and b: and the other figure would contain the results, clauses c,d,e of 

In the quantification of cytokines it is necessary to include the concentration of protein that was used to determine the different cytokines. 

Results Section

It is suggested to modify figure 1, because in this one the information of the biological model and results are being mixed, in this sense they could be separated, one figure could be only that of the model, that is to say clause a and b: and the other figure would contain the results, clauses c, d, e of 

Delete the information of line 234, "Figure 2: Colonic cytokine gene expression and histopathology scores differ in males and females exposed to RST or HCC conditions ", because it should be included as a figure caption and it is already duplicated.

Delete line 263 "Figure 3: Colonic cytokine levels differ in males and females exposed to RST or HCC conditions".

Discussion section

Could you please expand this information "Reduced body weight is a common consequence of cytokine-driven sickness in mice".

It is recommended that although the journal format indicates that the conclusion is optional, it would be convenient to include it to give more solidity to the manuscript.

Author Response

Reviewer 2 Comments:

Introduction section

  1. It is suggested to integrate information on the diversity of the microbiota described in lines 55-56.

Response: We have included a description of a study that has shown that enhanced colitis in ER-β deficient mice is related to the microbiome as a way to integrate information about the potential role of the microbiome in sex-based differences in colonic inflammation (lines 55-56).  However, in regards to diversity or specific bacterial taxa, there is a lack of consensus on what microbiome attributes are related to sex and are important for colonic inflammation. Mayneris et al. note that women have higher relative abundance of Akkermansia muciniphila and lower abundance of Bacteroidetes. Mueller et al. 2006 and Li et al. 2008 found greater abundance of  Bacteriodetes in men. Kim et al. provide a thorough summary of human studies investigating sex differences in the gut microbiota. An intriguing question is whether these sex-based differences in the microbiome contribute to alterations in intestinal inflammation. However, an assessment of the microbiome was beyond the scope of the current project.

Mayneris-Perxachs J, Arnoriaga-Rodríguez M, Luque-Córdoba D, et al. Gut microbiota steroid sexual dimorphism and its impact on gonadal steroids: influences of obesity and menopausal status. Microbiome. 2020;8(1):136. Published 2020 Sep 20. doi:10.1186/s40168-020-00913-x

Kim YS, Unno T, Kim BY, Park MS. Sex Differences in Gut Microbiota. World J Mens Health. 2020;38(1):48-60. doi:10.5534/wjmh.190009

  1. Could you indicate what you mean by evidence of the relationship between stress and intestinal inflammation (line 64-64), do you refer to some type of biomarkers? If so, could you integrate the information?

Response: We have previously found that stress leads to increases in inflammatory cytokines and chemokines, as well as colonic injury after challenge with a pathogen. This has now been included in the manuscript on lines 67-74.

  1. According to this information indicated in line 66, which phyla or genera of the intestinal microbiota were modified? 

Response: We did not assess the microbiome in the current study, and the effects of stress on microbiome composition is highly variable across experiments.  However, in previous studies by our group, we have consistently observed that restraint stress leads to reductions in alpha diversity and significant lower relative abundances of bacteria in the Lactobacillus genus. This information is now included in lines 69-71.

Maltz, R.M., et al., Prolonged restraint stressor exposure in outbred CD-1 mice impacts microbiota, colonic inflammation, and short chain fatty acids. PloS one, 2018. 13(5): p. e0196961.

Bailey et al., Stressor exposure disrupts commensal microbial populations in the intestines and leads to increased colonization by Citrobacter rodentium. Infection and Immunity, 2010. 78(4): 1509-1519.

Methodological section

  1. In the histology section, it is necessary to include information on the microscope with which the samples were observed, as well as to indicate on the basis of which author the classification of the damage was established.

Response: We have now included the magnification and the microscope type that was used for histological assessment (line145). In addition, we reference the scoring system that was used (lines 145-154).

  1. It is necessary to eliminate the information of line 134 because it should be a figure caption and this information is repeated:

Figure 1: Experimental design showing that T cell transfer is necessary for sickness-related weight changes". 

Response: We agree and have updated this.

  1. At the quantification of cytokines it is necessary to include the concentration of protein that was used to determine the different cytokines. 

Response: We expressed the cytokine concentration as pg per gram of colon tissue assessed as is often reported in the literature (see for example Shaler et al., 2021), but unfortunately, we did not quantify total protein levels.

Shaler C.R. et al., 2021. “Psychological stress impairs IL22-driven protective gut mucosal immunity against colonizing pathobionts.” Nature Communications, 12(1): 6664.

Results Section

  1. It is suggested to modify figure 1, because in this one the information of the biological model and results are being mixed, in this sense they could be separated, one figure could be only the one of the model, that is to say clause a and b: and the other figure would contain the results, clauses c,d,e

Response: We agree and have updated this in the manuscript.  

  1. In the quantification of cytokines, it is necessary to include the concentration of protein that was used to determine the different cytokines. 

Response: Please see response to Point 6 above.

  1. Delete the information of line 234, "Figure 2: Colonic cytokine gene expression and histopathology scores differ in males and females exposed to RST or HCC conditions ", because it should be included as a figure caption, and it is already duplicated.

Response: Thank you for this suggestion. This has been updated in the manuscript.

  1. Delete line 263 "Figure 3: Colonic cytokine levels differ in males and females exposed to RST or HCC conditions".

Response: Thank you for this suggestion. This has been updated in the manuscript.

Discussion section

  1. Could you please expand this information "Reduced body weight is a common consequence of cytokine-driven sickness in mice".

Response: This paragraph was expanded to include: “Reduced body weight is a common consequence of cytokine-driven sickness in mice [31]. The expansion of cytokines during sickness causes disruptions to homeostatic conditions, resulting in increased body temperature, increased sleep, reduced appetite and accompanying weight loss, as well as changes in metabolism [32]. Weight loss is a common symptom of the inflammatory response and has been used as an indicator of cytokine-driven sickness in mice [32, 33].” Lines 301-306

  1. It is recommended that although the journal format indicates that the conclusion is optional, it would be convenient to include it to give more solidity to the manuscript.

Response: This has been added to the manuscript.

Round 2

Reviewer 2 Report

Comments and Suggestions for Authors

Dear author, I appreciate your attention to the suggestions for changes and modifications requested. I wish you the best of success in future publications.